# Melatonin Inhibits Testosterone Synthesis in Rooster Leydig Cells by Targeting CXCL14 through miR-7481-3p

**DOI:** 10.3390/ijms242316552

**Published:** 2023-11-21

**Authors:** Haoran Xu, Jingxin Pu, Yunkun Teng, Qingyu Zhu, Lewei Guo, Jing Zhao, He Ding, Yi Fang, Xin Ma, Hongyu Liu, Jing Guo, Wenfa Lu, Jun Wang

**Affiliations:** 1Key Lab of Animal Production, Product Quality and Security, Ministry of Education, Jilin Agricultural University, Changchun 130118, China; haoranxu1197@163.com (H.X.); 18990907185@163.com (J.P.); yunkun1999@163.com (Y.T.); qingyuzhu1993@163.com (Q.Z.); guolewei163@163.com (L.G.); jlndzjing@126.com (J.Z.); dinghe1130@126.com (H.D.); 17743112414@163.com (Y.F.); maxin3202@163.com (X.M.); jlndlhy0133@163.com (H.L.); jguochn@163.com (J.G.); 2Jilin Province Engineering Laboratory for Ruminant Reproductive Biotechnology and Healthy Production, College of Animal Science and Technology, Jilin Agricultural University, Changchun 130118, China

**Keywords:** rooster, Leydig cells, melatonin, miR-7481-3p, CXCL14, testosterone, PI3K/AKT signal pathway

## Abstract

Melatonin has been proved to be involved in testosterone synthesis, but whether melatonin participates in testosterone synthesis by regulating miRNA in Leydig cells is still unclear. The purpose of this study is to clarify the mechanism of melatonin on Leydig cells testosterone synthesis from the perspective of miRNA. Our results showed that melatonin could significantly inhibit testosterone synthesis in rooster Leydig cells. miR-7481-3p and *CXCL14* were selected as the target of melatonin based on RNA-seq and miRNA sequencing. The results of dual-luciferase reporter assays showed that miR-7481-3p targeted the 3′-UTR of *CXCL14*. The overexpression of miR-7481-3p significantly inhibited the expression of CXCL14 and restored the inhibitory role of melatonin testosterone synthesis and the expression of StAR, CYP11A1, and 3β-HSD in rooster Leydig cells. Similarly, interference with *CXCL14* could reverse the inhibitory effect of melatonin on the level of testosterone synthesis and the expression of StAR, CYP11A1, and 3β-HSD in rooster Leydig cells. The RNA-seq results showed that melatonin could activate the PI3K/AKT signal pathway. Interference with *CXCL14* significantly inhibited the phosphorylation level of PI3K and AKT, and the inhibited PI3K/AKT signal pathway could reverse the inhibitory effect of *CXCL14* on testosterone synthesis and the expression of StAR, CYP11A1 and 3β-HSD in rooster Leydig cells. Our results indicated that melatonin inhibits testosterone synthesis by targeting miR-7481-3p/*CXCL14* and inhibiting the PI3K/AKT pathway.

## 1. Introduction

Testosterone is mainly produced by Leydig cells in male animals and plays an important role in spermatogenesis and sexual maturation [1]. It is well known that testosterone production requires the participation of steroid acute protein (StAR) [2], which plays an important role in the transportation of cholesterol to cytochrome P450 cholesterol side chain lyase (CYP11A1) [3]. When cholesterol reaches the inner membrane of mitochondria, it is converted into pregnenolone by the action of the rate-limiting enzyme P450scc of steroid synthesis, and finally into testosterone by the action of 3-hydroxysteroid dehydrogenase (3β-HSD) [4,5], but the mechanism of testosterone synthesis is not clear.

Melatonin is mainly produced by the pineal gland at the back of the cranial fossa. The synthesis of melatonin is regulated by photoperiod and has a circadian rhythm. Melatonin usually plays a biological role by specifically binding to melatonin membrane receptors and nuclear receptors, including membrane receptor 1 and membrane receptor 2 of the guanine nucleotide-binding regulatory protein-coupling receptor family, and melatonin nuclear receptor ROR orphan nuclear receptor. The synthesis of melatonin affects the hypothalamus–pituitary–testis axis, and promotes sexual activity and the synthesis of testosterone [6,7]. Previous studies have shown that melatonin promotes testosterone synthesis in sheep and goat Leydig cells [8,9]. Our previous studies identified the melatonin receptor types of roosters, and our results demonstrated that melatonin could bind to MEL1A and MEL1B to inhibit testosterone synthesis through the cAMP/PKA/CREB signaling pathway in rooster Leydig cells [10]. These results showed that melatonin is species-specific in regulating testosterone synthesis in different species.

MicroRNAs (miRNAs) are a kind of non-coding single-stranded RNA small molecule, which could regulate biological processes by inhibiting the expression of target genes [11]. At present, the functions of miRNA in different species have been widely reported [12,13,14]. Although previous studies have identified a variety of miRNAs that can regulate testosterone synthesis in goat, mouse, and pig [15,16,17,18], there was no information on the role of miRNA in the regulation of melatonin on testosterone synthesis. The aim of this study is to explore whether miRNA is involved in the inhibitory role of melatonin on testosterone synthesis and its mechanism in rooster Leydig cells.

## 2. Result

### 2.1. Effect of Melatonin on Testosterone Synthesis in Rooster Leydig Cells

In order to verify the effect of melatonin on rooster Leydig cells, the rooster Leydig cells were divided into the control group (CTR) and the melatonin treatment group (MT). Compared with CTR, the testosterone level, and the mRNA level of *StAR*, *CYP11A1*, and *3β-HSD* in MT significantly decreased (*p* < 0.05, Figure 1A–D).

### 2.2. Differentially Expressed miRNA with Melatonin Treatment in Rooster Leydig Cells

In order to investigate the potential role of miRNA in melatonin inhibiting testosterone synthesis in rooster Leydig cells, we analyzed the correlation between CTR and MT by PCA (Figure 2A). Compared with CTR, MT has 9 down-regulated differential miRNAs and 1 up-regulated differential miRNA (Figure 2B). We used the pheatmap software package in R language to conduct two-way cluster analysis on all miRNAs between CTR and MT (Figure 2C). 

### 2.3. Differentially Expressed mRNA with Melatonin Treatment in Rooster Leydig Cells

We used RNA-seq to select effective target genes. According to the expression level, we carried out PCA (principal component analysis) on CTR and MT (Figure 3A). Compared with CTR, MT has 262 down-regulated differential genes and 31 up-regulated differential genes (Figure 3B); meanwhile, clustering was based on the expression level of the same gene in different samples and the expression patterns of different genes in the same sample (Figure 3C). The KEGG analysis results suggested that melatonin could activate the PI3K/AKT signal pathway (Figure 3D). To determine whether the PI3K/AKT signaling pathway is involved in melatonin-induced testosterone synthesis, we treated rooster Leydig cells with melatonin for 36 h, and the results showed that melatonin could significantly increase the phosphorylation level of PI3K and AKT (Figure 3E,F).

### 2.4. CXCL14 Is the Direct Target Gene of miR-7481-3p in Rooster Leydig Cells

To explore the binding target of miR-7481-3p, we used the TargetScan database (www.targetscan.org, Date: 25 November 2021) to identify the possible downstream targets (Figure 4A), and we verified the expression levels of miR-7481-3p and *CXCL14*. The result suggested that *CXCL14* was identified as the target gene of miR-7481-3p (*p* < 0.01, Figure 4B,C). To further confirm this prediction, we used the psi-CHECK-2 vector containing the miR-7481-3p binding sites of *CXCL14* to perform the reporter assay, and the vector containing mutant binding sites was used as a control. The results showed that miR-7481-3p overexpression inhibited the activity of luciferase (*p* < 0.01, Figure 4D–F). Additionally, the expression of CXCL14 significantly reduced by overexpressing miR-7481-3p (*p* < 0.05, Figure 4G,H). These results indicated that *CXCL14* is the direct target of miR-7481-3p in rooster Leydig cells.

### 2.5. Effect of miR-7481-3p on Testosterone Synthesis in Rooster Leydig Cells

To better understand the functions of miR-7481-3p in the process of testosterone synthesis in rooster Leydig cells, we transfected rooster Leydig cells with miR-7481-3p mimics for 48 h (*p* < 0.05, Figure 5A), which was found to significantly increase the expression of StAR, CYP11A1, and 3β-HSD and the level of testosterone (*p* < 0.05, Figure 5B,C). These results indicated that miR-7481-3p could promote testosterone synthesis in rooster Leydig cells.

### 2.6. miR-7481-3p Inhibits Rooster Leydig Cell Testosterone Synthesis via CXCL14

In order to assess whether *CXCL14* mediates the effects of miR-7481-3p in rooster Leydig cells, we transfected miR-7481-3p mimics and interfered *CXCL14* for 48 h. The results showed that the level of testosterone increased significantly in rooster Leydig cells (*p* < 0.05, Figure 6A–D). These results indicated that *CXCL14* is the direct target of miR-7481-3p in rooster Leydig cells.

### 2.7. Melatonin Inhibits Testosterone Synthesis via miR-7481-3p in Rooster Leydig Cells

To prove whether miR-7481-3p was involved in melatonin inhibiting testosterone synthesis in rooster Leydig cells, we overexpressed miR-7481-3p for 12 h and then treated with melatonin for 36 h. The results showed that the overexpression of miR-7481-3p could also reverse the inhibitory effects of melatonin on testosterone levels and the expression of StAR, 3β-HSD, and CYP11A1 in rooster Leydig cells (*p* < 0.05, Figure 7A–G). These results indicated that the inhibitory effect of melatonin on testosterone was achieved by down-regulating the expression of miR-7481-3p.

### 2.8. Effect of CXCL14 on Testosterone Synthesis in Rooster Leydig Cells

To determine the effect of *CXCL14* on testosterone synthesis in rooster Leydig cells, we interfered *CXCL14* for 12 h to reduce its expression in Leydig cells (*p* < 0.05, Figure 8A). The results showed that interfering *CXCL14* could significantly promote testosterone levels in rooster Leydig cells, and significantly up-regulate the expression of StAR, CYP11A1, and 3β-HSD (*p* < 0.05, Figure 8B). These results suggested that *CXCL14* could inhibit testosterone synthesis in rooster Leydig cells.

### 2.9. Melatonin Inhibits Testosterone Synthesis via CXCL14 in Rooster Leydig Cells

To clarify the function of *CXCL14*, we interfered *CXCL14* for 12 h and treated rooster Leydig cells with melatonin for 36 h (*p* < 0.05, Figure 9A). The results showed that compared with the NC + MT groups, the expression of StAR, CYP11A1, and 3β-HSD and testosterone levels in si-CXCL14 + MT groups were significantly increased (*p* < 0.05, Figure 9B–G). These results demonstrated that melatonin could inhibit testosterone synthesis through *CXCL14* in rooster Leydig cells.

### 2.10. Melatonin Inhibits Testosterone Synthesis in Rooster Leydig Cells via the PI3K/AKT Pathway

Previous results show that melatonin can inhibit testosterone synthesis through miR-7481-3p/*CXCL14* in rooster Leydig cells, but it is not clear whether PI3K/AKT is involved. To determine whether the PI3K/AKT signaling pathway is involved in melatonin-induced testosterone synthesis, we treated rooster Leydig cells with or without 20 μM PI3K/AKT-IN-1, which is the inhibitor for PI3K/AKT, for 30 min, and then interfered *CXCL14* for 48 h, and found that interfering the expression of *CXCL14* and treating cells with melatonin could significantly inhibit the expression of p-PI3K and p-AKT (*p* < 0.05, Figure 10A,B).The results indicated that PI3K/AKT-IN-1 could reverse the inhibitory effect of *CXCL14* on the expression of StAR, CYP11A1, and 3β-HSD (*p* < 0.05, Figure 10C–E).

## 3. Discussion

Melatonin is a kind of neuroendocrine hormone secreted by the pineal gland, and it has biological functions such as regulating neuroendocrine cells [19], circadian rhythm [20], anti-inflammatory, and anti-oxidation [21]. Although our previous study indicated that melatonin could inhibit testosterone synthesis via the CREB signal pathway in rooster Leydig cells [10,22], whether melatonin regulates testosterone synthesis through miRNA was not clear. In the current study, our results showed that melatonin could inhibit testosterone synthesis by targeting miR-7481-3p/*CXCL14* and inhibiting the PI3K/AKT pathway.

miRNA is about 20–24 nucleotides in length and is a post-transcriptional regulator of gene expression [23,24]. miRNA plays a key role in participating in or regulating the synthesis of testosterone in Leydig cells [25]. Previous studies demonstrated that miR-1197-3p could promote the process of testosterone secretion in goat Leydig cells [15]. miR-140-5p can target SF-1 and ultimately inhibit the expression of enzymes related to testosterone synthesis in rat Leydig cells [16]. miR-150 negatively regulates the expression of STAR and steroid production in mouse Leydig cells [26]. In this study, it was indicated for the first time that miR-7481-3p could inhibit the synthesis of testosterone by inhibiting the expression of *CXCL14* in rooster Leydig cells. Meanwhile, our study here proved for the first time that melatonin could inhibit testosterone synthesis by down-regulating miR-7481-3p expression in Leydig cells. In addition, in our small RNA sequencing results, other differentially expressed miRNAs were also selected, such as miR-449a, miR-451, miR-6552-5p, miR-449b-5p, miR-126-5p, miR-101-2-5p, miR-6575-5p, and miR-7459-3p. We speculate that these miRNAs may be potential targets involved in the process of testosterone synthesis, which needs further research to prove.

*CXCL14* is a kind of pleiotropic cytokine which belongs to the C-X-C motif chemokine ligand family [27,28,29]. A previous study demonstrated that *CXCL14* could activate the p38/JNK signaling pathway, and up-regulate the expression of StAR, which is a key enzyme in progesterone synthesis, and promote progesterone synthesis through cyclic adenosine monophosphate response element binding protein (CREB) phosphorylation in a polycystic ovary syndrome (PCOS) human model [30]. We speculate that *CXCL14* may be involved in the synthesis of steroid hormones. Our RNA-seq results showed that in the process of melatonin inhibiting testosterone synthesis in rooster Leydig cells, the expression of *CXCL14* changed significantly; further results show that *CXCL14* could inhibit testosterone synthesis in rooster Leydig cells, and *CXCL14* could be involved in mediating the inhibitory effect of melatonin on testosterone synthesis in rooster Leydig cells. In addition, we proved that CXCL14 is the direct target of miR-7481-3p. Our results showed that melatonin could up-regulate *CXCL14* and inhibit the synthesis of testosterone in rooster Leydig cells. To our knowledge, this is the first direct evidence that *CXCL14* is involved in testosterone synthesis in Leydig cells.

The function of the PI3K/AKT signaling pathway has been widely reported. The PI3K/AKT signaling pathway can regulate cell proliferation [31], survival [32], cell migration [33], growth [34], metabolism [35], and apoptosis [36]. Previous studies showed that ER oxidoreduclin 1α could activate the PI3K/AKT/mTOR signaling pathway and promote testosterone synthesis mediated by hCG in mice Leydig cells [37]. Additionally, in a rat PCOS model, miR-18b-5p could target PTEN, promote the activation of the PI3K/AKT signaling pathway, and improve PCOS [38]. Meanwhile, melatonin could activate the PI3K/AKT signaling pathway to stimulate the expression of StAR and progesterone synthesis in bovine Theca cells [39]. Melatonin could inhibit miR-15a-5p and activate Stat3 and the PI3K-Akt-mTOR pathway, and block the autophagy of granulosa cells [40]. However, there was little evidence that melatonin could regulate the process of testosterone synthesis in Leydig cells through the PI3K/AKT signaling pathway. In the current study, our RNA-seq results indicated that melatonin can activate the PI3K/AKT signaling pathway and may be a potential downstream target of *CXCL14*. Moreover, our results indicated that interfering *CXCL14* could significantly inhibit the phosphorylation of PI3K and AKT, and then treatment with PI3K/AKT-IN-1 could significantly promote testosterone synthesis in rooster Leydig cells. To our knowledge, our results proved for the first time that melatonin could target miR-7481-3p/*CXCL14* to activate the PI3K/AKT signaling pathway and inhibit testosterone synthesis in rooster Leydig cells. Unfortunately, this study only proved in vitro that melatonin could activate the phosphorylation level of PI3K and AKT and inhibit testosterone synthesis by targeting miR-7481-3p/*CXCL14* in rooster Leydig cells, and the above process has not confirmed whether miR-7481-3p could regulate testosterone synthesis in vivo.

## 4. Materials and Methods

### 4.1. Animals and Cells Isolation

All animal experiments were in accordance with the regulations of the Animal Management Committee of Jilin Agricultural University (20230216002). Thirty experimental animals were selected from sexually mature roosters (22 weeks old) at a rooster farm in Dehui, Jilin, and euthanasia was performed after feeding them for one week. The rooster testes were collected and kept in normal saline at 37 °C. Primary Leydig cells were isolated with 1 ng/mL collagenase II (Sigma, Burlington, MA, USA). After the addition of DMEM/F12 (Gibco, Waltham, MA, USA), 10% fetal bovine serum (Sangon Biotech, Shanghai, China), and a small amount of penicillin-streptomycin solution, the cells were cultured in an incubator at 37 °C with 5% CO_2_ for 24 h to ensure the number of cells per mL reached 2.5 million. The extraction and purification processes of Leydig cells are consistent with our previous research [10,22]. All experiments were carried out in the Animal Products Quality and Safety Laboratory of Jilin Agricultural University.

### 4.2. Rooster Leydig Cell Treatment

According to our previous research method [10,22], the primary rooster Leydig cells were treated with or without 1 ng/mL melatonin (M5250-1G, Sigma-Aldrich, St. Louis, MO, USA), and transfected or not transfected with miR-7481-3p mimics or si-CXCL14 (Genepharma, Suzhou, China). Based on the manufacturer’s standards, each 2μM of mimics or inhibitor plasmid was combined with 20 uL of Lipofectamine^TM^ 2000 (Sigma, Burlington, MA, USA), and the Leydig cells were incubated in an incubator with 37 °C, 5% CO_2_. After 6 h, DMEM/F12 medium was replaced, and all transfected fragments were synthesized by Suzhou Genepharma. (Table 1). According to PI3K/AKT-in-1 manufacturer’s instructions, the roosters Leydig cells were treated with PI3K/AKT-in-1 (10 μM, MedChemExpress, Monmouth Junction, NJ, USA) for 30 min, and then treated with 1 ng/mL melatonin for 36 h. In all experiments the number of Leydig cells was 5 million (2 mL).

### 4.3. Testosterone Enzyme-Linked Immunosorbent Assay (ELISA)

According to our previous research [22], the rooster Leydig cells supernatant was collected, and the testosterone level was detected according to the instructions of the kit (Meimian industrial Co., Ltd., Yancheng, Jiangsu, China). The enzyme was incubated at 37 °C to allow the enzyme to function and bind to the substrate, and enzyme-labeled reagents, A and B display solutions, were added in turn, according to the instructions. The experimental results were observed with MicroplateReader (Thermo, New York, NY, USA) at 450 nm light wave. Again, each treatment and experimental grouping was repeated three or more times.

### 4.4. Quantitative Real-time Polymerase Chain Reaction (qRT-PCR) Analysis

The total RNA of the primary Leydig cells was extracted using the Trizol method (Invitrogen, Carlsbad, CA, USA), and then 1 μg of RNA was reverse-transcribed into cDNA according to the instructions of the RT kit (TaKaRa, Kyoto, Japan). All qRT-PCR processes were carried out on an ice box. The expressions of StAR, CYP11A1, 3β-HSD, miR-7481-3p, and CXCL14 were detected on the Agilent Strata Gene Mx3005P system (Santa Clara, CA, USA) with SYBR-Green master mix reagent (TaKaRa, Kyoto, Japan) (Table 2), and the relative 2^−ΔΔCt^ algorithm was finally used to analyze their relative quantities. The expression level was compared with the expression level of β-actin (ACTB, Sangon Biotech, Shanghai, China).

### 4.5. Western Blot Analysis

The total protein was isolated from Leydig cells with RIPA cell lysis buffer (Beyotime, Shanghai, China) and analyzed for protein concentration with the enhanced BCA protein assay kit. A total of 20 μg of protein in different groups was separated by 10% sodium dodecyl sulfate-polyacrylamide gel electrophoresis (SDS-PAGE). The separated proteins were transferred to NC membranes by semi-dry blotting (Millipore, Burlington, MA, USA). All membranes were incubated in blocking buffer (LI-COR, USA) for 90 min, primary antibodies StAR (1:1000, bs-3570R, Bioss Antibodies, Beijing, China), CYP11A1 (1:1000, 13363-1-AP, ProteinTech, Chicago, IL, USA), 3β-HSD (1:1000, DF3653, Affinity, Jiangsu, China), PI3K (1:1000, 4249S, CellSignalingTechnology, Boston, MA, USA), AKT (1:1000, 9272S, CellSignalingTechnology, Boston, MA, USA), p-PI3K (1:1000, 4228S, CellSignalingTechnology, Boston, MA, USA), p-AKT (1:1000, 4060S, CellSignalingTechnology, Boston, MA, USA), and β-actin (1:10000, 60008-1-1g, ProteinTech, Chicago, IL, USA) were added, and the membranes were placed overnight at 4 °C. All membranes were washed with TBST and incubated with a secondary antibody (1:10000, ProteinTech, Chicago, IL, USA). Finally, the bands were observed with enhanced luminescence reagent, and the gray value was calculated. The gray value of the control group was set to 100%.

### 4.6. Small RNA Sequencing and Transcriptome Sequencing Data Analysis

The total RNA from rooster Leydig cells was extracted with Trizol reagent (TaKaRa, Kyoto, Japan). The quality and integrity of the extracted RNA was assessed on an Agilent 2100 Bioanalyzer. After the small RNA-seq library was constructed, PCR amplification was performed to enrich the library. The Agilent 2100 Bioanalyzer was used to perform a quality inspection of the library with the Agilent High Sensitivity DNA Kit. The Quant-iT PicoGreen dsDNA Assay Kit was used to quantitate the library, and the optimal loading amount was selected for sequencing on the Illumina platform.

After obtaining the total RNA of the rooster Leydig cells, the NovaSeq platform was used, and the original data were obtained using on-machine sequencing with reference to the NCBI genome database (www.ncbi.nlm.nih.gov (accessed on 12 October 2023), Gallus gallus. GRCg6a. dna. top level. fa). The data were filtered, and the filtered high-quality sequences were compared with those on the reference genome of the species. The samples were further subjected to expression difference analysis, enrichment analysis, and cluster analysis based on the comparison results. All raw data for this study have been submitted to the NCBI Gene Expression Omnibus database.

### 4.7. Dual-Luciferase Reporter Assay

The 30 non-coding regions of *CXCL14* were synthesized by Shanghai Sangon Biotech Co., Ltd. China, including the predicted binding site and the modified mutation site of miR-7481-3p. The sequence was cloned into the psi-CHECK-2 vector (Promega, Madison, WI, USA) between the XhoI and NotI sites, and expression was regulated by the SV40 promoter. A total of 10 μL DMEM was mixed with 3′-UTRwt and 3′-UTRmuta target plasmid and 5 pmol negative control (A); then, 10 μL DMEM was mixed with 0.3 μL transfection reagent (B) and placed at room temperature for 5 min. After the AB solution was mixed, it was placed at room temperature for 20 min. Cells were harvested after 48 h in the incubator and detected with the Promega Dual-Luciferase system.

### 4.8. Statistical Analysis

Student’s t-test was used to assess the significance of differences between groups. All data were statistically analyzed using SPSS, and all graphs were plotted using GraphPad Prism Version 5.0 (GraphPad Software, San Diego, CA, USA). All experimental values met statistical significance (*p* < 0.05). At least three independent experiments were performed for each outcome (*n* = 3).

## 5. Conclusions

In this study, we demonstrated that melatonin could inhibit testosterone synthesis in rooster Leydig cells by targeting miR-7481-3p to inhibit the expression of CXCL14 and by activating the PI3K/AKT signal pathway.

## Figures and Tables

**Figure 1 ijms-24-16552-f001:**
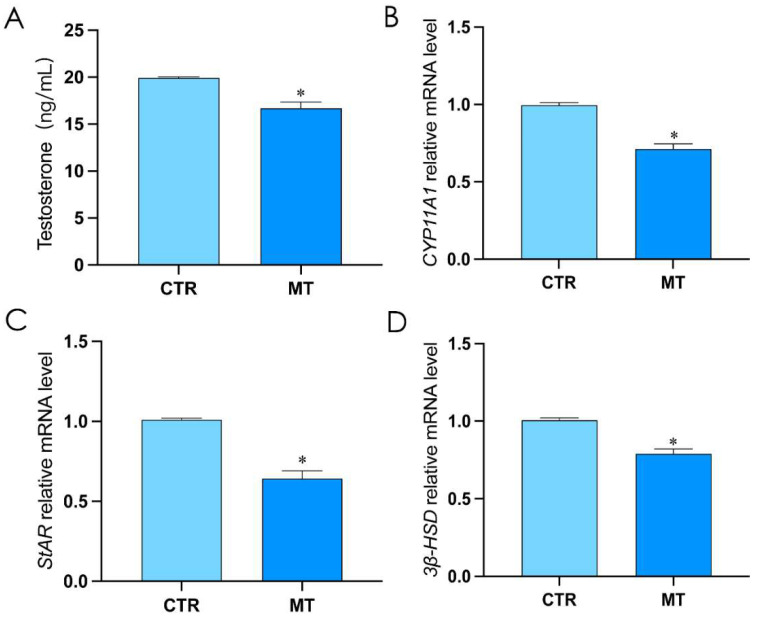
Melatonin inhibits testosterone synthesis in rooster Leydig cells. (**A**) The level of testosterone in rooster Leydig cells treated with melatonin for 36 h. (**B**–**D**) The relative mRNA expression of StAR, CYP11A1, and 3β-HSD in rooster Leydig cells treated with melatonin for 36 h. All data are presented as the mean ± SD (*n* = 3); * *p* < 0.05.

**Figure 2 ijms-24-16552-f002:**
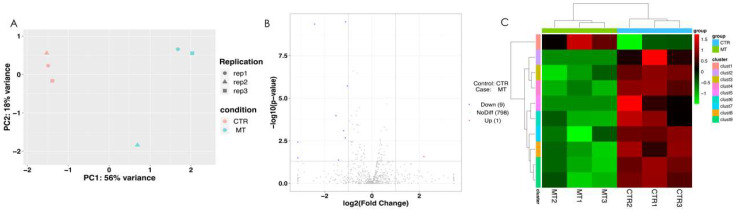
Differentially expressed miRNA under melatonin treatment in rooster Leydig cells. (**A**) The analysis of PCA between CTR and MT. (**B**) Differentially expressed miRNA between CTR and MT. (**C**) Cluster analysis of all miRNAs between CTR and MT. All data were presented as the mean ± SD (*n* = 3).

**Figure 3 ijms-24-16552-f003:**
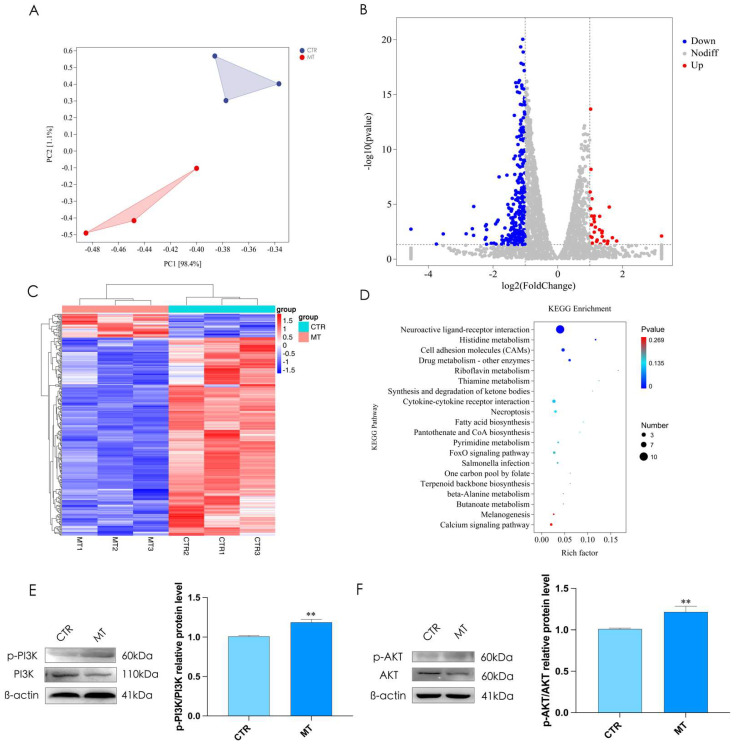
Differentially expressed mRNA under melatonin treatment in rooster Leydig cells. (**A**) Principal component analysis for CTR and MT. (**B**) Differentially expressed mRNA between the CTR group and the MT group. (**C**) Cluster analysis of all mRNAs between CTR and MT. (**D**) KEGG analysis between CTR and MT. (**E**,**F**) The phosphorylation level of PI3K and AKT between CTR and MT. All data are presented as the mean ± SD (*n* = 3), ** *p* < 0.01.

**Figure 4 ijms-24-16552-f004:**
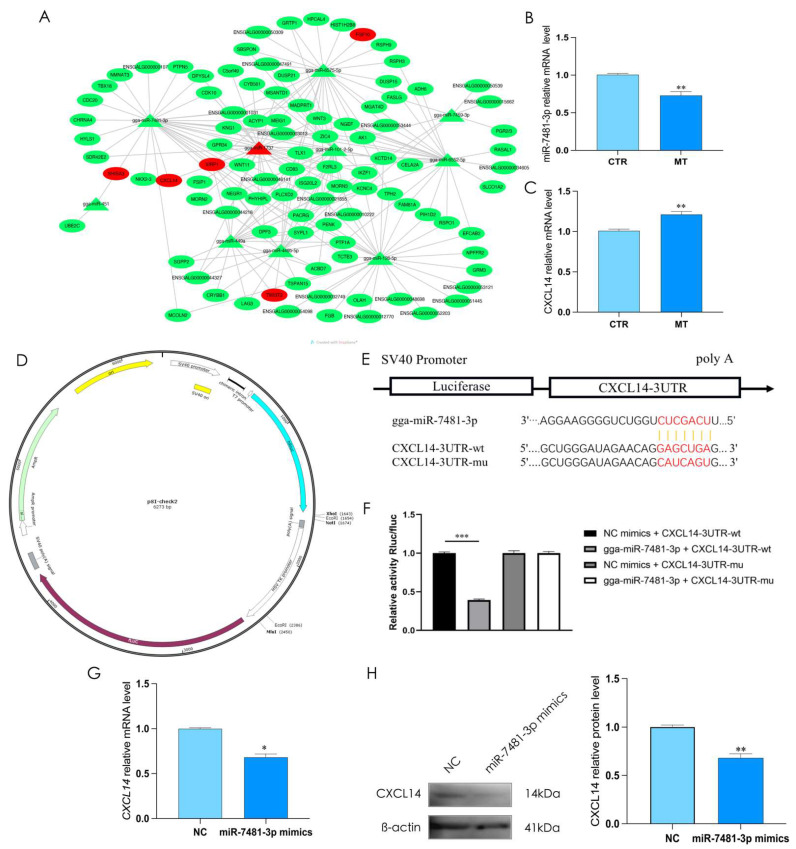
*CXCL14* is the direct target of miR-7481-3p. (**A**) Predicted potential targets of miR-7481-3p. (**B**,**C**) The relative mRNA expression of miR-7481-3p and *CXCL14* in rooster Leydig cells. (**D**) Predicted miR-7481-3p binding sites in the *CXCL14* 3′-UTR. (**E**) The luciferase reporter vector fused with *CXCL14* 3′-UTR WT or its MUT was constructed. (**F**) Dual luciferase assay on luciferase activity in different treatment groups. (**G**,**H**) Over-expressed miR-7481-3p to detect the relative mRNA and protein expression of CXCL14 in rooster Leydig cells. All data are presented as the mean ± SD (*n* = 3), * *p* < 0.05, ** *p* < 0.01 and *** *p* < 0.001.

**Figure 5 ijms-24-16552-f005:**
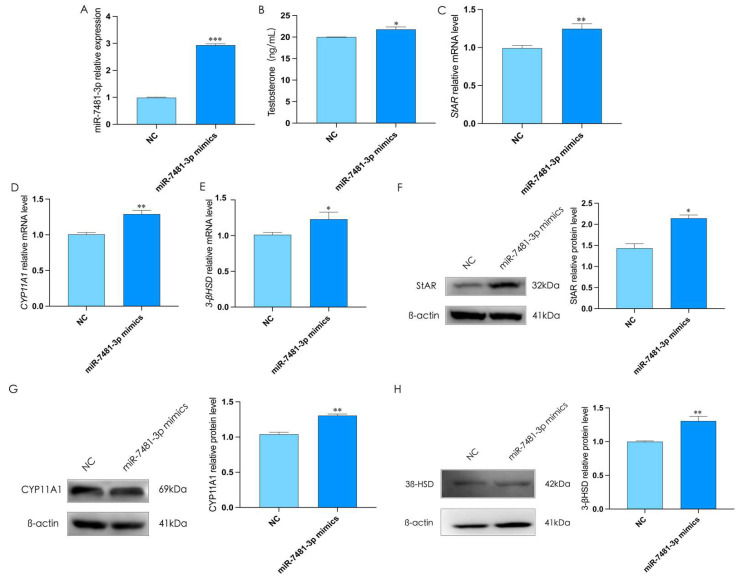
Overexpression of miR-7481-3p promotes testosterone synthesis in rooster Leydig cells. (**A**) Treatment with miR-7481-3p mimics for 48 h increases the relative mRNA expression of miR-7481-3p in roosters Leydig cells. (**B**) The level of testosterone in rooster Leydig cells treated with miR-7481-3p mimics for 48 h. (**C**–**H**) Treatment with miR-7481-3p mimics for 48 h increases the relative mRNA and protein expression of StAR, CYP11A1, and 3β-HSD in rooster Leydig cells. All data are presented as the mean ± SD (*n* = 3), * *p* < 0.05, ** *p* < 0.01 and *** *p* < 0.001.

**Figure 6 ijms-24-16552-f006:**
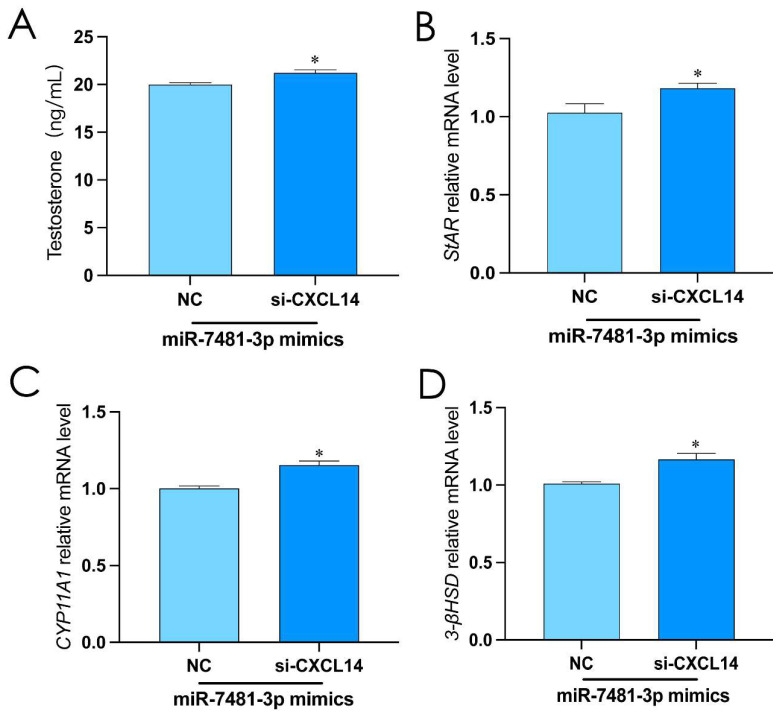
miR-7481-3p inhibits rooster Leydig cell testosterone synthesis via *CXCL14*. (**A**) Over-expressed miR-7481-3p and interfered *CXCL14* for 48 h to detect the level of testosterone in rooster Leydig cells. (**B**–**D**) Over-expressed miR-7481-3p and interfered *CXCL14* to detect the relative mRNA expression of StAR, CYP11A1, and 3β-HSD in roosters Leydig cells. All data are presented as the mean ± SD (*n* = 3), * *p* < 0.05.

**Figure 7 ijms-24-16552-f007:**
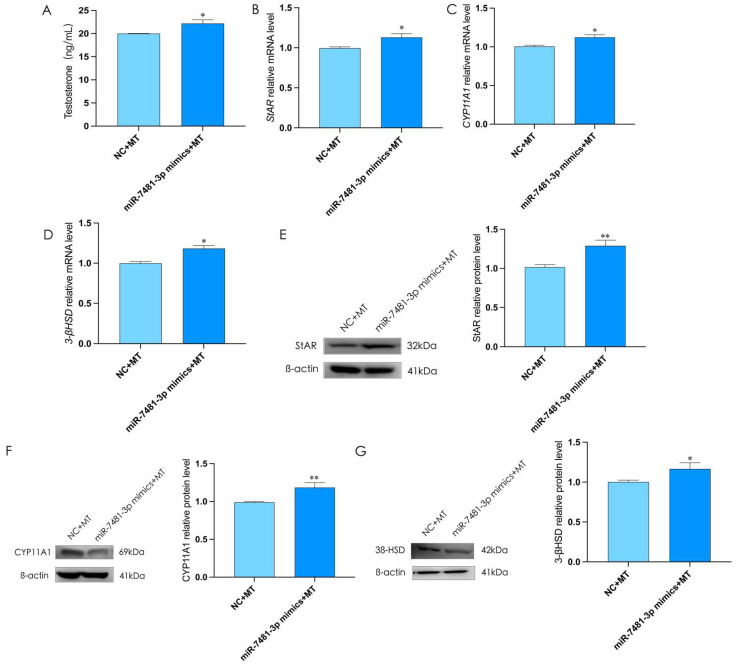
The overexpression of miR-7481-3p weakened the inhibitory effect of melatonin on testosterone synthesis in rooster Leydig cells. (**A**) The level of testosterone in rooster Leydig cells treated with miR-7481-3p mimics and melatonin for 48 h. (**B**–**G**) Treatment with miR-7481-3p mimics and melatonin for 48 h increased the relative mRNA and protein expression of StAR, CYP11A1, and 3β-HSD in rooster Leydig cells. All data are presented as the mean ± SD (*n* = 3), * *p* < 0.05 and ** *p* < 0.01.

**Figure 8 ijms-24-16552-f008:**
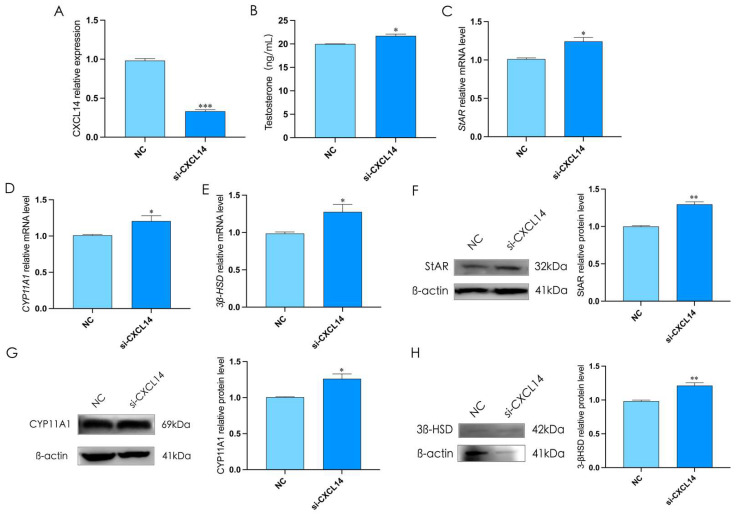
The interference of *CXCL14* promotes testosterone synthesis in rooster Leydig cells. (**A**) The interference of *CXCL14* for 48 h decreased the relative mRNA expression of *CXCL14* in rooster Leydig cells. (**B**) The interference of *CXCL14* for 48 h increased the level of testosterone in rooster Leydig cells. (**C**–**H**) The interference of *CXCL14* for 48 h increased the relative mRNA and protein expression of StAR, CYP11A1, and 3β-HSD in rooster Leydig cells. All data are presented as the mean ± SD (*n* = 3), * *p* < 0.05, ** *p* < 0.01 and *** *p* < 0.001.

**Figure 9 ijms-24-16552-f009:**
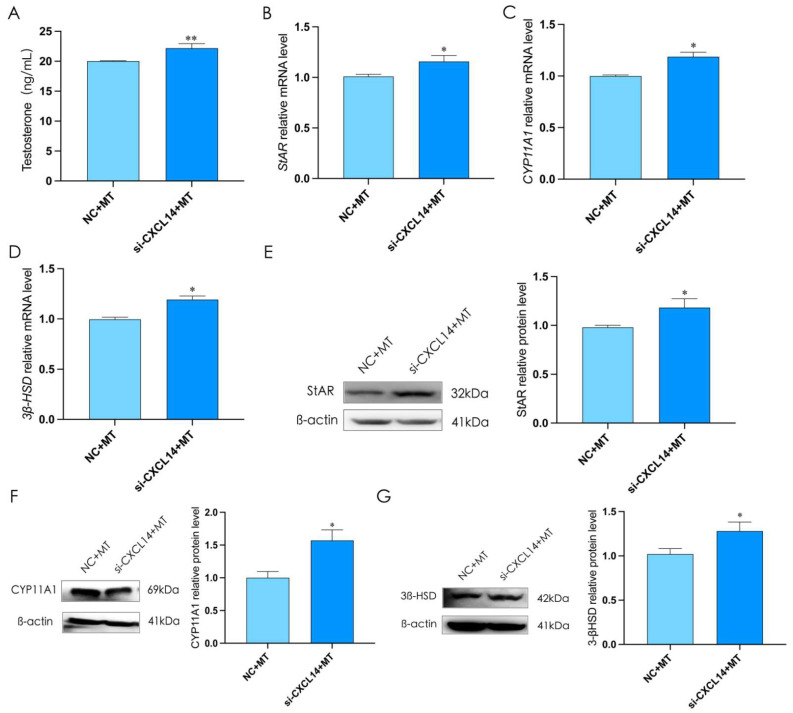
The interference of *CXCL14* weakened the inhibitory effect of melatonin on testosterone synthesis in rooster Leydig cells. (**A**) The interference of *CXCL14* and treatment with melatonin for 48 h increased the level of testosterone in rooster Leydig cells. (**B**–**G**) The interference of *CXCL14* and treatment with melatonin for 48 h increased the the relative mRNA and protein expression of StAR, CYP11A1, and 3β-HSD in rooster Leydig cells. All data are presented as the mean ± SD (*n* = 3), * *p* < 0.05 and ** *p* < 0.01.

**Figure 10 ijms-24-16552-f010:**
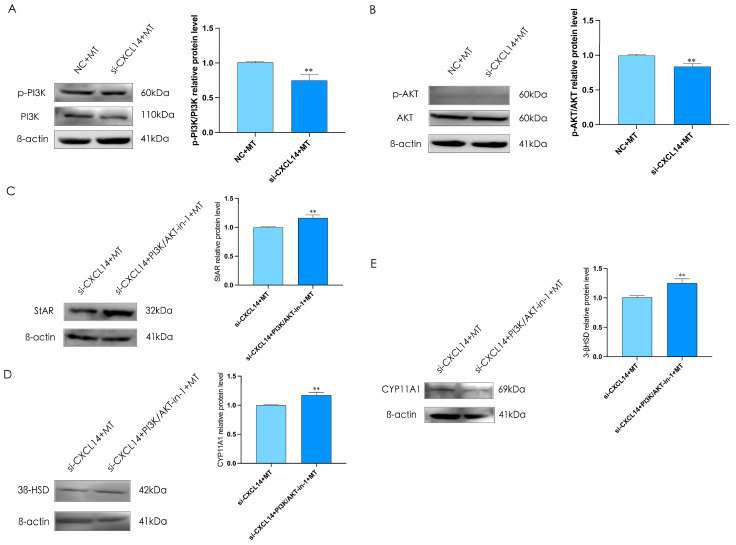
The PI3K/AKT pathway mediates melatonin inhibition of testosterone synthesis in rooster Leydig cells. (**A**,**B**) The interference of *CXCL14* and treatment with melatonin for 36 h decreased the relative protein expression of p-PI3K, and p-AKT in rooster Leydig cells. (**C**–**E**) The treatment with melatonin and interference of *CXCL14* and PI3K/AKT-IN-1 for 48 h increased the relative protein expression of StAR, CYP11A1, and 3β-HSD. All data are presented as the mean ± SD (*n* = 3), ** *p* < 0.01.

**Table 1 ijms-24-16552-t001:** Sequences of NC, miR-7481-3p mimics, and si-CXCL14.

Name	Sequence (5′-3′)
NC	Sense: UUCUCCGAACGUGUCACGUTT
	Antisense: ACGUGACACGUUCGGAGAATT
miR-7481-3p mimics	Sense: UUCAGCUCUGGUCUGGGGAAGGA
	Antisense: CUUCCCCAGACCAGAGCUGAAUU
si-CXCL14	Sense: GAUUCUCUAACGUACGGAATT
	Antisense: UUCCGUACGUUAGAGAAUCTT

**Table 2 ijms-24-16552-t002:** Primers used for the qRT-PCR analysis.

Gene	Primer Sequence(5′→3′)	Annealing Temperatures/°C
*StAR*	F: GCGGACAACGGAGACAAAGT R: TGATCCACCACCACCTCCAG	60
*CYP11A1*	F: GCCACGCTCTTCAAGTCAGA R: GGTAGTCACGGTATGCCACC	59
*3β-HSD*	F: TAAGCGTGTTATCATCTC R: CTGGGGAAACAGCAACAGCAG	54
miR-7481-3p	F: CGTTCAGCTCTGGTCTGGG R: AGTGCAGGGTCCGAGGTATT	60
*CXCL14*	F: GTGACCCTGTGGACGAAAGTGAG R: ACCCTGCCCTTCTCCTTCCATAC	60
*ACTB*	F: TAAGCGTGTTATCATCTC R: GGGACTTGTCATATTTCT	52–60

## Data Availability

The data that support the findings of this study are available from the corresponding author upon reasonable request.

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
