# Peer review of "Melatonin Inhibits Testosterone Synthesis in Rooster Leydig Cells by Targeting CXCL14 through miR-7481-3p"

_ijms, 2023, doi:10.3390/ijms242316552_

Round 1

Reviewer 1 Report

Comments and Suggestions for Authors

The experimental approaches were well planned, but many initial basic definition were missing in figure 

1) there is no information for approve of animal experiments (IACUC), and there is no information how many rooster testes were used for this experiments, and how old etc.. no detail information of animals. 

2) Author described that how to isolated rooster Leydig cell but there is no evidence that isolated Leydig cell is pure Leydig cell... pleased add results which define the charateristics of this cells. IHC with leydig cell marker 

This is most important stuff for trust of these results. 

3) In material and method, qRT-PCR. did you buy primer? line 92 described the expression levels was compared with the expression levels of beta actin (ACTB, Sangon Biotch, Shanhai) is this company for synthesis of primer?

4) As you know, there is several different type of melatonin recepotor (1A, 1B, MT3 etc..)

First auhtor need to identify the which receptor is positive to rooster leydig cells by immunostaining 

5) how many repeat thie experiment (Biological and experimental?) 

Author Response

Thank you for your letter and for the reviewers’ comments concerning our manuscript entitled “Paper Title” (ID: ijms-2688087). Those comments are all valuable and very helpful for revising and improving our paper, as well as the important guiding significance to our research. We have studied comments carefully and have made correction which we hope meet with approval. We posted two word file, including updated manuscript and reply to reviewer 1 on the website.

Reviewer 2 Report

Comments and Suggestions for Authors

Overall, the topic is interesting; however, there are some suggestions that authors should consider before making progress:
In lines 41–46, the authors should expand on the pharmaceutical features of melatonin, such as cell permeability.
Also, in the introduction, the authors should add information about the endogenous production of melatonin in the testis and the expression of melatonin receptors in the testis of different species, particularly the studied species.
The physiological function of melatonin in the male reproductive system and the fact that melatonin and testosterone share similar circadian rhythms should be added to the introduction.
The animal age range and sample size should be provided in the method section.
The authors should mention the reason for the 24-hour cell culture period. Was that the time required for the 80% confluency or something else?
The dissolvent of melatonin and PI3K/AKT-in-1 should be added to the method section.
In line 64, the density or concentration of cells (the number of cells per mL) should be clarified for both culture and treatment. How many cells/mL were exposed to 10 μM of PI3K/AKT-in-1 or 1 ng/mL of melatonin?
In lines 68–76, the authors should explain how they selected the PI3K/AKT and melatonin concentrations and treatment times. Did they do dose-response tests? So, it should be explained in detail.
Or they chose the concentration and exposure time based on the previous studies. If yes, they should cite the reference(s).
In lines 79–84, more detailed information should be provided about the ELISA assessment.
Also, was testosterone measured in the culture medium of Leydig cells? If not, why is it only measured in the cells?
In the method section, protocols for all tests should be cited.
Please provide the device name of all tests in the methodology. Some of them are missing, such as the ELISA technique.
There are some typographical errors, e.g., table 2: “annealing”. The entire manuscript should be checked for that.
In the result section, the expression panels for beta-actin, negative control (if possible), and positive control (if possible) should be shown in figures 3E and F. The same comment applies to figures 4 G, 5 G, and H; figure 7 E, F, G; figure 8 F,G,H; figure 9 E, F, G; and figure 10.
Also, better representative images for western blotting should be provided. The figures are blurry, particularly the p-AKT figure. The same comment applies to figure 4G.

The discussion is too short, so it needs to be improved. It should be expanded on the possible underlying molecular mechanisms and important pathways that might be involved in the results the authors obtained in the current study. Please also compare and contrast the obtained results with previous studies.

The limitations of the current study should be noted.

Author Response

Thank you for your letter and for the reviewers’ comments concerning our manuscript entitled “Paper Title” (ID: ijms-2688087). Those comments are all valuable and very helpful for revising and improving our paper, as well as the important guiding significance to our research. We have studied comments carefully and have made correction which we hope meet with approval. We posted two word file, including updated manuscript and reply to reviewer 2 on the website.

Round 2

Reviewer 1 Report

Comments and Suggestions for Authors

The paper has improved after the first round of reviews. 

Reviewer 2 Report

Comments and Suggestions for Authors

Thanks to the authors for their efforts. The manuscript has been improved.